# Manipulating and monitoring nanoparticles in micellar thin film superstructures

Jan Bart ten Hove [1,2], Fijs W.B. van Leeuwen[1,2] & Aldrik H. Velders [1,2]

Understanding the dynamics of discrete self-assembled structures under influence of external triggers is of interest to harvest the potential of nano- and mesoscale materials. In particular, controlling the hierarchical organization of (macro)molecular and nanoparticle building blocks in monolayer superstructures is of paramount importance for tuning properties and characteristics. Here we show how the electron beam in cryo-transmission electron microscopy can be exploited to induce and follow local migration of building blocks and global migration of micellar aggregates inside micrometer-sized superstructures. We employ stroboscopic exposure to heat up and convert the vitrified superstructure into a liquid-like thin film under cryogenic conditions, resulting in controlled evaporation of water that finally leads to rupture of the micelle-containing superstructure. Micelle-embedded nanoparticles prove a powerful tool to study the complex hierarchically built-up superstructures, and to visualize both global movement of individual dendrimicelles and local migration of nanoparticles inside the micellar core during the exposure series.

[1] Laboratory of BioNanoTechnology, Wageningen University & Research, Axis, Bornse Weilanden 9, 6708 WG Wageningen, The Netherlands. [2] Interventional Molecular Imaging Laboratory, Department of Radiology, Leiden University Medical Centre, Leiden 2333 ZA, The Netherlands. Correspondence and requests for materials should be addressed to A.H.V. (email: aldrik.velders@wur.nl)

The assembly of well-defined building blocks into super-structures is of widespread interest because of the emerging properties of such materials, in, for example, catalysis[1] storage[2], optics[3], and drug delivery[4,5]. Although three-dimensional self-assembled materials are studied most, such as gels and metal-organic frameworks, their two-dimensional counterparts are of interest too, and include monolayers of atomic, molecular, self-assembled, and colloidal building blocks[6–13]. Among the strategies reported to engineer two-dimensional self-assembled materials, interfaces have proven to provide a versatile template to do so[14–19]. Recently, we showed how micrometer-sized, freestanding dendrimicelle monolayer superstructures could be made from nanometer-sized building blocks, bridging three levels of hierarchy during the templated self-assembly and organization[20,21]. We hypothesized that the micelle-embedded nanoparticles could be used to study the rupture dynamics of such superstructures following electron beam-induced heating during cryo-transmission electron microscopy (cryoTEM).

Here, we use stroboscopic cryoTEM to observe movements inside monolayers of nanoparticle-containing superstructures, revealing the nanoscale processes that take place just before and during the thin film rupture. The nanoparticles have a dualistic purpose: In addition to revealing dendrimicelle characteristics such as aggregation numbers and core size, their contrast also allows for use as tracers to follow the dynamics of the super-structure under extended electron irradiation[22]. Moreover, the gold nanoparticles—with their high electron density—prove to homogeneously increase the local heating of the superstructures under irradiation, leading to the concomitant evaporation of water in the thin film and finally to the rupture of the liquid-like remaining superstructure. Interestingly, the electron beam allows manipulation of the building blocks of the aggregates at the lower nanometer length scales, including the removal of the gold par-ticles from the dendrimer host.

## Results

**Dendrimicelle and superstructure formation**. Our strategy, as depicted in Fig. 1a, is as follows: dendrimer-encapsulated gold nanoparticles (AuDENs) were synthesized inside positively charged sixth-generation (G6) PAMAM dendrimers, yielding $Au_{128}$DENs. Coacervation with anionic-neutral $pMAA_{64}pEO_{885}$ block copolymers yielded highly stable dendrimicelles (Supple-mentary Figures 1–3) as indicated by Dynamic Light Scattering (DLS), while cryoTEM proved these are indeed well-defined dendrimicelles (Supplementary Figure 4). The dendrimicelles host an average of $30 \pm 10$ nanoparticles in the micelle core, which—based on a 1:1 association of positive and negative charges—implies that the average micelle molecular weight is ~8 MDa. These dendrimicelles organize into superstructures, located in the thinnest part of the thin water film spanning across the TEM grid hole (Supplementary Figure 5). The formation of these superstructures is the result of the complex interplay between the poly(ethyleneglycol) of the dendrimicelle corona and the tem-plating effect of the biconcave thin film[23–26].

**Electron beam-induced thin film superstructure rupture**. Next, we studied the effect of prolonged electron beam irradiation on these self-organized superstructures of self-assembled micelles (Fig. 1b). During cryo-electron microscopy, inelastic scattering of electrons by the sample causes evaporation of the vitreous ice layer, which can result in destruction of the sample and of the thin water film in which the sample is embedded[27–30]. Fig. 2 shows that prolonged electron beam exposure results in the rupture of the dendrimicelle superstructure. The combination of

the exposure time (2 s) and the velocity at which the rupture propagated results in the rupture being captured as streaks, similar to motion blur observed in e.g., long exposure photo-graphy. The observed streaks originate from the gold nano-particles inside the dendrimicelles; their electron density clearly provides enough contrast to visualize the rupture of the thin film, while the macromolecular building blocks of the dendrimicelles are not providing enough contrast. The red arrows in Fig. 2 are drawn parallel to the gold nanoparticle streaks, and extrapolation indicates these to originate from the center of the superstructure. The center of this image lacks visible streaks, despite den-drimicelles being present before rupture. The rupture therefore likely started right before the cryoTEM micrograph was recorded, or the contrast provided by the micelle-embedded nanoparticles was attenuated over too large an area, and effectively merged with the background signal. Therefore, we can only obtain a minimum value for the rupture propagation speed, as determined from the nanoparticle streaks visible. Assuming the particles moved for the full 2 s of exposure time, the length of the streaks in the center of the figure indicates that dendrimicelles moved at an average speed of ~50 nm s$^{-1}$ during the exposure; the dendrimicelles at the edges of the superstructure appear to have moved at an average speed of ~ 25 nm s$^{-1}$. These observations point to the rupture starting from the center of the superstructure, followed by a radially outward propagation of the rupture front. Supplementary Figures 6 and 7 show more examples of dendrimicelle super-structure ruptures, supporting this hypothesis. Because of the visual similarities between this nanoscale rupture and an explo-sion (i.e., the sudden, destructive, outward migration), we coined this rupture nanoexplosion. Most thin film rupture studies regard thin soap films, where formation of many smaller bubbles during rupture is observed[31], but contrary to these aerosol studies, in our case we do not observe formation of smaller bubbles in the rupture process.

**Stroboscopic exposure of dendrimicelle superstructures**. Uniquely, to investigate the processes that take place inside the thin film just before and during the nanoexplosion, we use the micelle-embedded nanoparticles as tracer to follow the rupture of thin film under stroboscopic exposure (i.e., by taking pictures successively)[30]. Figure 3a shows a (~4 × 4 μm) cryoTEM micro-graph, in which a 2.4 μm circular TEM grid hole is present. Inside this grid hole, a ~500 nm diameter dendrimicelle superstructure is residing in the center. Figure 3d shows the dendrimicelle super-structure, acquired at higher magnification and at higher electron flux (~41$e^-$ Å$^{-2}$ per micrograph, acquired with a ~2 μm electron beam spot size). After a series of 14 micrographs and a total electron exposure dose of ~600 e$^-$ Å$^{-2}$ (see Supplementary Fig-ure 8 for the full series), the dendrimicelle superstructure explo-ded, leaving a hole behind roughly similar in size to the original dendrimicelle superstructure (Fig. 3b, f). The stroboscopic TEM micrographs allow for detailed analysis. Figure 3d–f show three representative figures of the 14 cryoTEM micrographs taken during the stroboscopic exposure, with Fig. 3d showing the first image of the exposure series, in which the individual den-drimicelle cores in the superstructure are clearly visible. In the last frame before the nanoexplosion (Fig. 3e), the superstructure area appears as a smooth film, which suggests that the majority of the water that was present has evaporated, leaving a thin polymeric film behind with groups of DENs originating from the original micelle cores. The next frame, Fig. 3f, shows the superstructure just at the final stage of the nanoexplosion. Figure 3b shows that after the nanoexplosion the exposed area (i.e., the area exposed to the ~2 μm circular electron beam) appears lighter-colored. This corresponds to an increased electron transmission, implying the

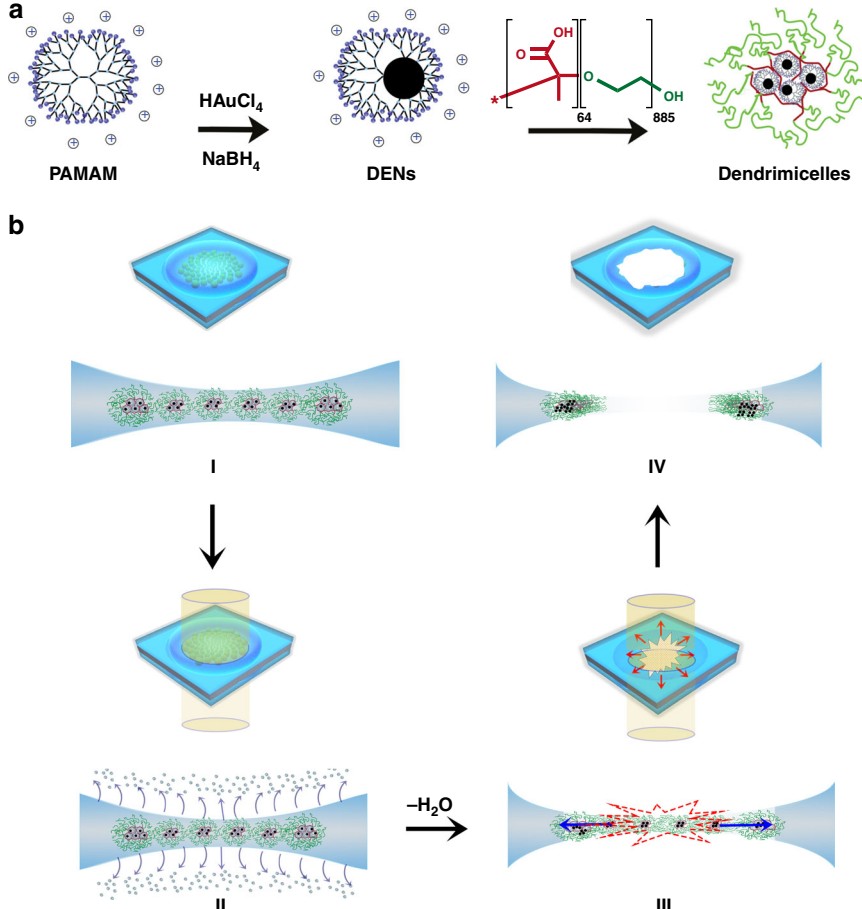

**Fig. 1** Synthesis and manipulation of monolayer dendrimicelle superstructures. **a** Complexation and reduction of Au(III) ions inside PAMAM dendrimers provides dendrimer-encapsulated gold nanoparticles (AuDENs). Mixing of these AuDENs with a negative-neutral pMAA$_{64}$PEO$_{885}$ block copolymer results in ~50 nm dendrimicelles. **b** For cryoTEM, micrometer-sized grid holes are filled with an aqueous thin dendrimicelle film, with the biconcave shape of this layer acting as a template for the organization of dendrimicelles into (sub-) micrometer-sized monolayer superstructures (I). Exposure of the superstructure to an electron beam results in evaporation of water (II). Prolonged exposure results in the rupture of the superstructure, with the embedded AuNPs revealing local movements of dendrimicelle building blocks as well as the global migration of dendrimicelles during the nanoexplosion (III). After the nanoexplosion, a hole similar in size to the superstructure is left behind (IV)

exposure series resulted in a thinning of the vitrified thin film, also outside of the superstructure area. Plotting the average image intensity over the dendrimicelle superstructure (Fig. 3g and Supplementary Figure 9) indicates that during the exposure series the image transmission steadily increased, corresponding to a thinning of the film. Supplementary Figure 10 suggests a linear relationship between the image intensity and the exposure dose, corresponding to the well-defined evaporation of water during each exposure. The observed thinning and the homogeneous intensity of the superstructure layer just before nanoexplosion suggest a phase transition occurs with the thin vitreous ice layer formed during sample preparation, converting it into an even thinner, liquid-like, layer of organic, nanoparticle containing material.

The image intensity plot (Fig. 3c) confirms that before the stroboscopic exposure series (in red, corresponding to Fig. 3a), the dendrimicelle superstructure is located at the thinnest part of the biconcave thin film. After the stroboscopic exposure series (in blue, corresponding to Fig. 3b), the average image intensity has increased over the exposed area, implying that the electron beam transmission has increased, corresponding to a decreased thickness of the thin film (see also Supplementary Figures 9 and 10). These observations support our claim that the stroboscopic exposure

resulted in a thinning of the film and indicate that specifically the exposed part of the vitreous ice layer decreased in thickness, with the thickness of the unexposed part of the vitreous water film remaining virtually unchanged. The increased electron scattering of the micelle-embedded gold nanoparticles likely serves to aggravate heating and evaporation of water. Inelastic scattering (i.e., of electrons) transfers about 20 eV to the sample[32], and dissipated as heat, this can increase the local specimen temperature and leading to evaporation of the irradiated area[29,30]. Moreover, the hole present after the nanoexplosion allows to estimate the amount of water that evaporated during the exposure series. By assuming that the dendrimicelle superstructure was a 50-nm-thick monolayer before exposure (i.e., the average diameter of a dendrimicelle), the image intensity over the induced hole (Fig. 3c) can be converted to a thickness in nm. This translates to a thickness of the remaining vitreous water film of about 10 nm at the edge of the hole after the nanoexplosion, suggesting that a ~3 nm thick layer of water evaporated over the irradiated area during every exposure. Taking the height of a single layer of water to be the size of a water molecule (~0.3 nm), this infers that the water evaporation rate was on the order of ~10 layers of water molecules per exposure. Given the electron exposure rate of ~41 $e^-$ Å$^{-2}$ per exposure, this corresponds to a dose of ~30 electrons required for the evaporation

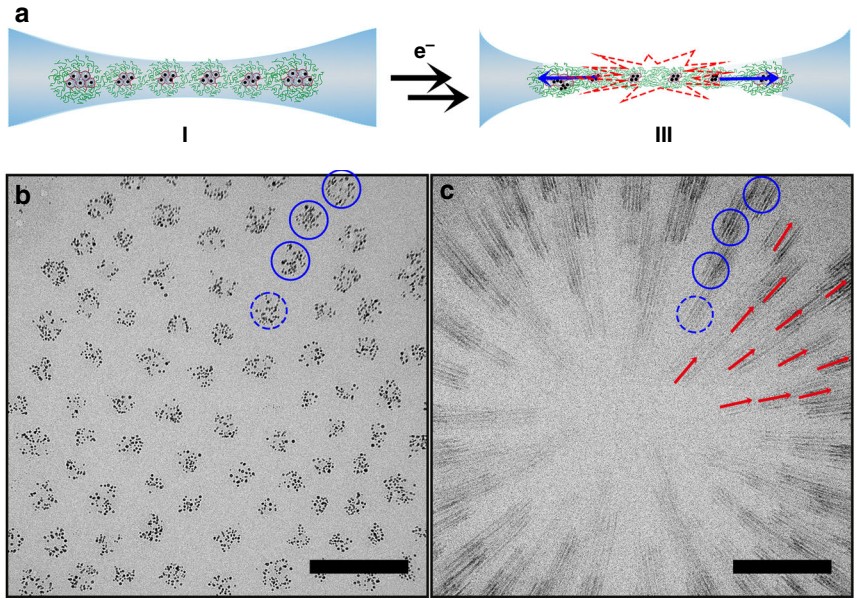

**Fig. 2** Electron beam irradiation-induced rupture of monolayer dendrimicelle superstructures. **a** Prolonged exposure of G6-Au$_{128}$-based dendrimicelles results in the rupture of the thin film in which they are embedded. CryoTEM micrographs of the dendrimicelles superstructure before (**b**) and during the nanoexplosion (**c**). Because of the relatively long exposure time and the contrast provided by the gold nanoparticles, the rupture process is captured as streaks. The rupture starts from the center of the dendrimicelle superstructures (as indicated with an asterisk) and radially propagates outwards. The blue solid circles are drawn as a guide to the eye around four dendrimicelles before and after rupture. The red arrows are drawn as a guide to the eye to show the directionality of the nanoparticle streaks. Scale bars are 100 nm

of each molecule of water. Figure 3h–j shows enlarged sections of the cryoTEM micrographs shown in Fig. 3d–f, highlighting an individual dendrimicelle, with the dendrimicelle core area indicated in Fig. 3h by the black circle surrounding the gold nanoparticles. Surprisingly, Figure 3i seems to indicate that right before the nanoexplosion, the gold nanoparticles are located closer together than they were in Fig. 3h. This observation suggests that the dendrimicelle core shrunk during the electron irradiation. Measurements of the dendrimicelle core area between these figures confirmed that the core area shrunk by about 40% during the stroboscopic exposure (Supplementary Figure 11, and see Supplementary Movie 1). Since the interior of PAMAM dendrimers[33] as well as the cores of complex coacervate micelles[34] are known to contain substantial amounts of water, this contraction is likely due to the evaporation of dendrimer- and micelle-entrapped water. Alternatively or in addition, it also might be that the gold nanoparticles simply have been pushed out of the shrinking dendrimer cavities, and therefore appear closer together, possibly surrounded by dendrimer material and hence forming so-called dendrimer-stabilized nanoparticles. To investigate the global migration of dendrimicelles during the extended electron beam exposure, we analyzed the image series using Particle Image Velocimetry (PIV). This technique divides the original image in smaller parts, and—based on the image shift between subsequent frames—calculates the corresponding velocity vector. The obtained velocity plots depicted in Supplementary Figure 12 show the radially-outward migration of dendrimicelles prior to the nanoexplosion. Likely, this dendrimicelle migration induces a stress on the dendrimicelle superstructure, finally leading to the nanoexplosion. We do note, however, that the limited time-domain resolution restricts quantification of the dendrimicelle migration dynamics.

**Local migration inside individual dendrimicelles**. To study the apparent dendrimicelle core shrinkage in more detail, we prepared a dendrimicelle sample from ninth-generation (G9)

Au$_{1024}$DENs at off-stoichiometric mixing ratio. Not only does the increased dendrimer generation allow for the encapsulation of larger nanoparticles inside, it also leads to fewer dendrimers per dendrimicelles enlarging contrast and facilitating visualization of the contents of the micelle core[21]. Figure 4a shows the obtained dendrimicelle superstructure before stroboscopic exposure, with the dendrimicelles in the center of the superstructure containing ~9 nanoparticles per dendrimicelle. The enlarged cryoTEM micrographs in Fig. 4c–e show the same location at various time points during the exposure series, with the whole exposure series shown in Supplementary Figure 13 and Supplementary Movie 2. These figures show the appearance of gas bubbles (Fig. 4c) that consecutively disappear (Fig. 4d), followed by the shrinkage of the dendrimicelles core (i.e., the clustering of the AuNPs; Fig. 4e). Plotting the average image intensity against total exposure dose (see Supplementary Figures 14 and 15) shows again the controlled decrease in thickness of the irradiated area. Plotting the dendrimicelle core area, i.e., as defined by the circle encompassing the nanoparticles, versus the exposure dose (Supplementary Figure 16) indicates that the average core volume decreased from ~10$^4$ to ~2 × 10$^3$ nm$^3$. As the volume of a hydrated generation nine dendrimer is already about ~10$^3$ nm$^3$, this observation indicates that the dendrimicelle core has not just only shrunk significantly during the exposure series. In fact, the appearance of gas bubbles during TEM irradiation (see Supplementary Figure 17) has been attributed to the evolution of hydrogen gas derived from the radiolysis of organic matter[27–29]. The apparent shrinking of the micelle core and the gas bubbles suggest that the dendrimicelle core has been—at least partially—damaged; making it plausible that the gold nanoparticles have ceased to be dendrimer-*encapsulated*-nanoparticles. The grouping of the gold nanoparticles in this process might be related to the magnetic properties attributed to gold nanoparticles[35]. The PIV analysis results, as shown in Supplementary Figure 18, suggest anisotropy in the global migration direction of the dendrimicelles prior to the

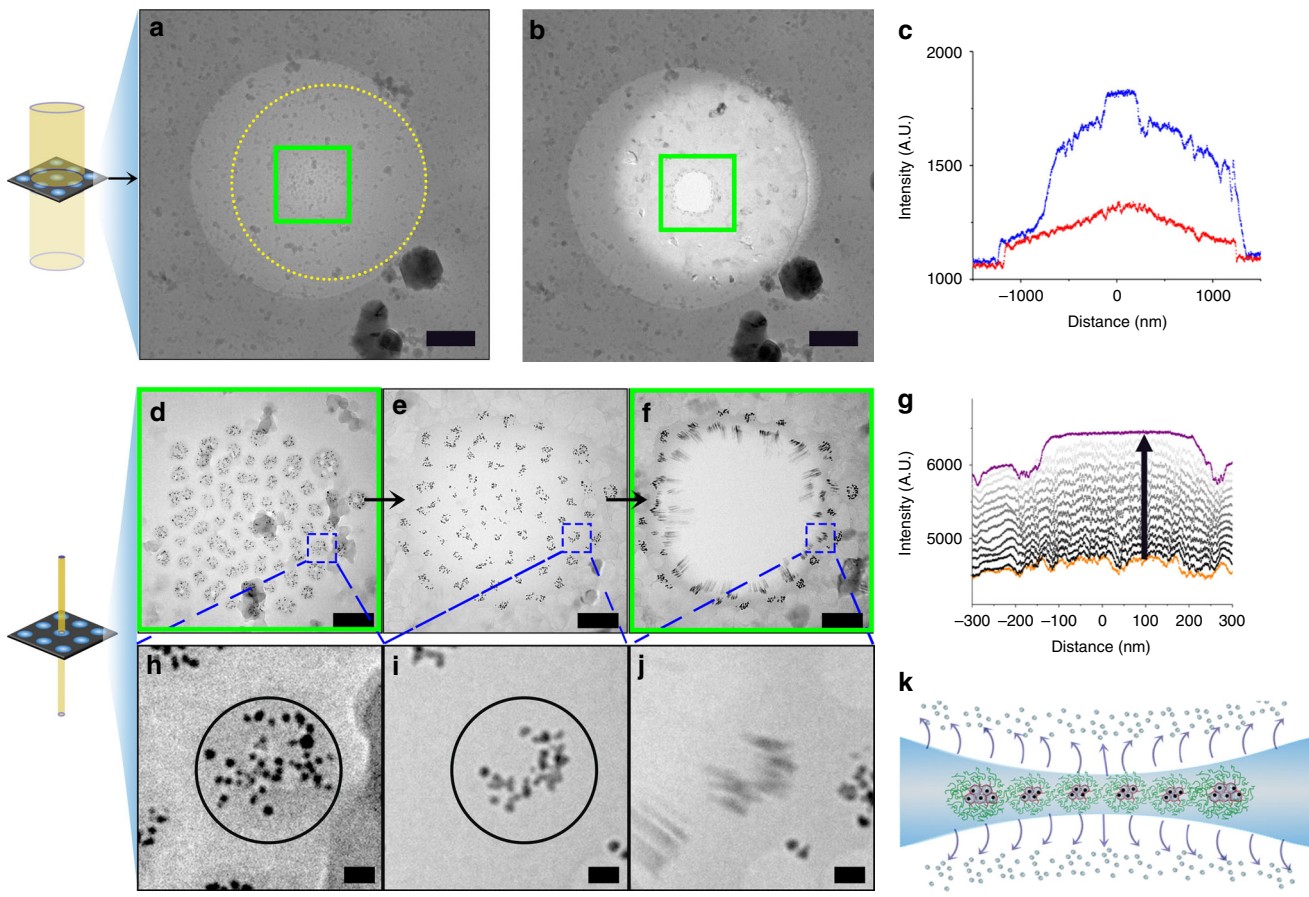

**Fig. 3** The effect of stroboscopic electron exposure on a nanoparticle-containing dendrimicelle superstructure. **a** CryoTEM micrograph of a generation-six based dendrimicelle superstructure obtained at low electron flux. **b** CryoTEM micrograph obtained after high electron flux stroboscopic exposure series. The high electron flux irradiation resulted in the evaporation of water from the irradiated area, decreasing the local thickness of the water film, corresponding to a lighter color. The yellow dotted circle in **a** indicates the size of the electron beam used for the exposure series, and the green solid square in **a** and **e** show the area of the superstructure that is captured by the microscope's camera during the high electron flux stroboscopic irradiation. **c** The image intensity plot over the cryoTEM gridholes shown in **a** (in red) and **b** (in blue) shows that the stroboscopic exposure series resulted in an increase in the image intensity values, corresponding to a decrease in thickness. **d**–**f** CryoTEM micrographs during high electron flux stroboscopic irradiation, showing: **d** the dendrimicelle superstructure at the start of irradiation, **e** just before the nanoexplosion, and **f** right after the nanoexplosion.The dashed blue boxes in **d**–**f** indicate the superstructure area that has been enlarged in panels **h**–**j**. **g** Image intensity plot during the high flux stroboscopic irradiation series. The graph shows that exposure yielded a gradual increase in image intensity values, corresponding to a decrease of the film thickness in the irradiated area. **h**–**j** Enlarged versions of the corresponding micrographs as shown in **d**–**f**, respectively, showing a single dendrimicelle during the exposure series. The black, solid circle in **h, i** illustrate the original dendrimicelle core area. **k** Cartoon illustrating the induced water evaporation during the stroboscopic exposure. Scale bars are 500 nm in **a**, **b**, 100 nm in **d**–**f**, and 10 nm in **h**–**j**

nanoexplosion, which was also observed before (e.g., Supplementary Figure 6). This anisotropy is likely due to small differences in the thickness of the dendrimicelle superstructure, caused by cryoTEM sample preparation.

The processes that take place during the clustering of the nanoparticles and the final nanoexplosion include the following: Extended electron beam irradiation results in heating of the exposed area, with the density of the nanoparticle-containing dendrimicelle core likely providing additional local heating. The increase in temperature of the vitreous ice layer results in the evaporation of water, from both the water layer and from the dendrimicelle core (Fig. 4f); in addition, the partial radiolysis of hydrocarbons inside the dendrimicelle core generates gas bubbles. At the start of the exposure series, the dendrimicelle corona is still visible, and there is contrast between dendrimicelle coronas. Just before the nanoexplosion, however, this contrast disappears, and a smooth, dendrimicelle superstructure is observed (as shown in Figs. 3e and 4e). Since both PAMAM as well as the complex coacervate micelle cores are hydrated[33,34], evaporation of water

could cause the shrinkage of the core as well as the dendrimers (Fig. 4g), applying stress on the thin film. Further evaporation of water likely amplifies the stress inside the superstructure, leading to the rupture starting from the thinnest point, the superstructure center. Apart from observing a radially outward-directed movement of the micelles during the nanoexplosion and the shrinkage of the dendrimicelle prior to the nanoexplosion, we also observed global migration of dendrimicelles (Supplementary Figure 19) and even the local migration and clustering of gold nanoparticles inside the dendrimicelle cores, as shown in Supplementary Figure 20. These results show that the dendrimicelle superstructures become viscous, liquid-like, thin films rather than the solid vitreous structures as obtained from the cryoTEM sample preparation.

## Discussion

In summary, we here exploited dendrimer-encapsulated gold nanoparticles, embedded inside dendrimicelle superstructures, to

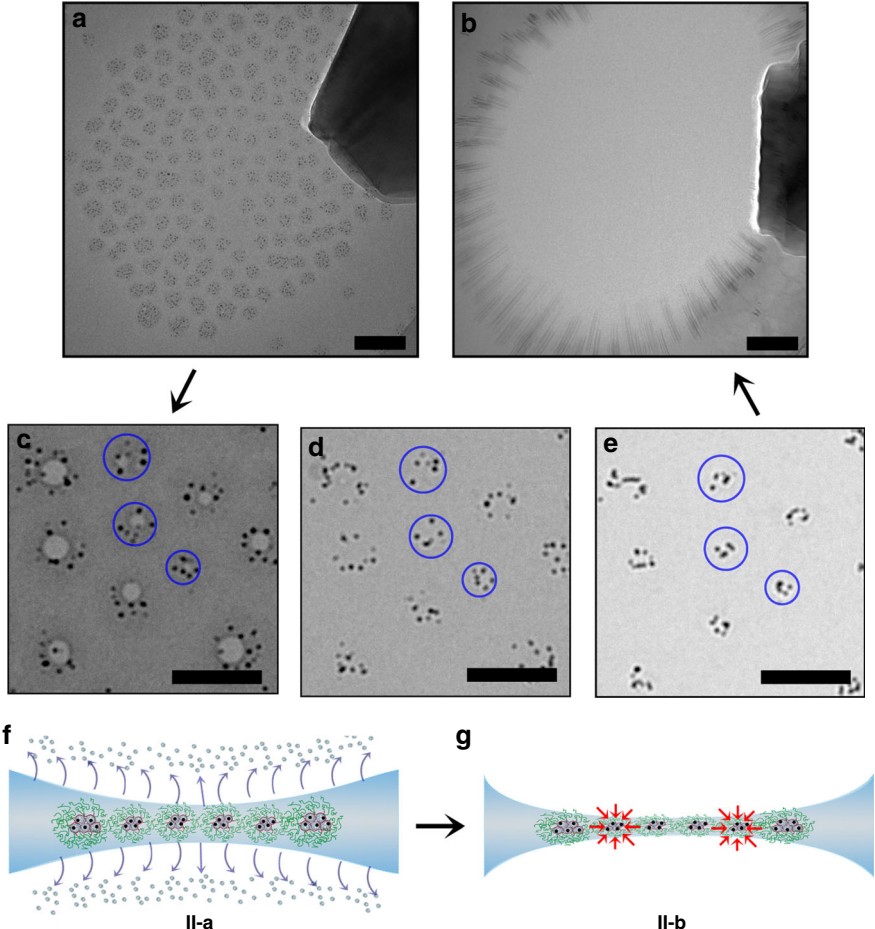

**Fig. 4** Local and global migration in dendrimicelle superstructures under electron beam irradiation. **a** CryoTEM micrograph of a dendrimicelle superstructure obtained from dendrimicelles encapsulating ninth-generation PAMAM dendrimers in their core, with an $Au_{1024}$ nanoparticle encapsulated inside every dendrimer. **b** CryoTEM micrograph after the electron beam-induced nanoexplosion of the dendrimicelle superstructure. **c–e** Enlarged sections of the superstructure at various time points during the electron beam irradiation, showing that upon prolonged irradiation, the contrast provided by the dendrimicelle core is reduced, and the gold nanoparticles appear closer together. The blue circles in **c–e** represent the original dendrimicelle core area, and are drawn to visualize the observed dendrimicelle core shrinkage. **f**, **g** Schematic illustration of the different stages during the nanoexplosion. CryoTEM sample preparation provides a dendrimicelle superstructure located at the thinnest part of the biconcave water film. **f** Extensive electron irradiation results in the evaporation of water from the biconcave thin film, leading to the formation of a freestanding polymer film in the center of the grid hole, in which the dendrimicelle cores seems to have shrunk, but more likely the gold particles have been excreted from the dendrimer voids and group together (**g**). The scale bars are 100 nm

study local and global processes that take place before and during electron beam-induced rupture of vitrified thin films containing organized superstructures of well-defined, nanoparticle-containing, micelles. The micelle-embedded nanoparticles act as tracer by providing contrast in the cryoTEM micrographs and allow for visualizing the superstructure rupture, which we coined nanoexplosion. The gold nanoparticles not only provide information on the global migration patterns during the nanoexplosion, they also visualize migration of individual dendrimicelles and show local movement of the gold nanoparticles inside individual dendrimicelles. For future work, the highly modular dendrimicelles allow to systematically investigate relevant parameters for thin film rupture at the nanoscale. It not only allows for varying the type, size, and number of nanoparticles per dendrimicelle, but also allows for changing the number of nanoparticles-containing dendrimicelles and the dendrimicelle polydispersity per superstructure. Gold nanoparticle assemblies are of interest for optical[36] and sensor[37] applications, as well as for metamaterials[38]. This study opens up thin film physics at the

nanometer scale, of relevance for novel materials that are being developed for, e.g., medicinal, catalytic, magnetic, or optical properties.

## Methods
**Materials**. Sixth-generation, amine-terminated polyamidoamine (PAMAM) dendrimers, (3-(*N*-morpholino)-propanesulfonic acid) (MOPS), $NaBH_4$, 1 M NaOH and 1 M HCl solutions were obtained from Sigma Aldrich. $pMAA_{64}$-b-$PEO_{885}$ (Mw/Mn = 1.15) was obtained from Polymer Sources Inc., Canada and used as 5 mM solution based on carboxylic acid content. $HAuCl_4.3H_2O$ was obtained from TCI. Ninth-generation, amine-terminated PAMAM was obtained from Dendritech, Inc., USA.

**Apparatus**. Dynamic Light Scattering was done on a Malvern Zetasizer Nano S equipped with a laser operating at 633 nm. For cryoTEM, samples were cast on Quantifoil R2/2 grids or 400 mesh Holey Carbon grids. After blotting, samples were plunged into liquid ethane using a Vitrobot system (FEI Company). Samples were then imaged at ~90–100 K in a JEOL 2100 TEM operating at 200 kV or JEOL 1400Plus TEM operating at 120 kV. The electron dose was calibrated using the microscope's software at the start of a cryoTEM session. Sample grids for electron microscopy were obtained from Electron Microscopy Sciences (EMS, Hatfield, PA,

USA) and were rendered hydrophilic using a plasma cleaning setup (15 s at $10^{-1}$ Torr). TEM image analyses were done using custom MATLAB scripts (The MathWorks Inc., USA) and FIJI (https://fiji.sc/).

**Dendrimer encapsulated nanoparticles.** G6-Au$_{128}$DENS were made following established protocols[39]. Shortly, 50 μL (35 nmol) of 5 wt% PAMAM G6-NH$_2$ in methanol was transferred to a 5 mL vial and the solvent was evaporated under reduced pressure. Next, 2 mL of water was added to dissolve the PAMAM and the pH was adjusted to 3 using 1 M HCl, after which 128 molar equivalents of Au$^{3+}$ to PAMAM were added. The resulting solution was then stirred for 20 min, after which 44 μL of a 1 M solution of NaBH$_4$ in 0.3 M NaOH (10 molar equivalents to Au$^{3+}$) were added. This resulted in the reduction of Au$^{3+}$ to AuDENs, indicated by the change from colorless to a dark brown solution within seconds after addition. Following reduction, the pH was set to 7 using HCl and the AuDENs were stored at 4 °C. Generation 9 AuDENs were made in a similar way, using 1024 equivalents of Au$^{3+}$ to PAMAM.

**Dendrimer encapsulated nanoparticles in micelles.** To obtain dendrimicelles under charge stoichiometric conditions, 20 μL of 2.89 mM dendrimer solution (charge concentration, 57 nmol positive charge based on surface groups) was dissolved in 149 μL water and 20 μL of 0.2 M MOPS buffer at pH 7.0 was added. Then, 11 μL pMAA$_{64}$-b-PEO$_{885}$ (55 nmol based on −COOH) was added and the sample was sonicated for 2 min. Generation 9-based dendrimicelles were made using 1.5 charge equivalents of pMAA$_{64}$PEO$_{885}$ to G9$_{1024}$DENS to decrease the average number of nanoparticles per dendrimicelle. Samples were left to equilibrate for at least one day before characterization using cryoTEM.

**Particle image velocimetry.** Particle Image Velocimetry analysis was done using a custom written script by cross-correlation the same region in subsequent frames. Velocity vectors were calculated from the determined image shifts and were threshold based on the original image intensity and number of neighboring vectors. Outlying vectors were replaced by their neighborhood average[40].

**Code availability.** The PIV Matlab code used during the current study is available from the corresponding author on reasonable request.

## Data availability
The datasets generated during and/or analyzed during the current study are available from the corresponding author on reasonable request.

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

## Acknowledgements
J.B.tH. and A.H.V. thank NWO for financial support of the MONALISA project 717.013.006. The research leading to these results has received funding from the

European Research Council (ERC) under the European Union's Seventh Framework Program FP7/2007-2013 (Grant No. 2012-306890) and the Horizon 2020 research and innovation programme under the Marie Sklodowska-Curie grant agreement No. 642192. We thank the Wageningen Electron Microscopy Centre for their support with the cryoTEM measurements.

## Author contributions

J.B.t.H. acquired the data, J.B.t.H. and A.H.V. analyzed the data and wrote the manuscript, J.B.t.H., F.W.B.v.L. and A.H.V. were involved with the design of the study.

## Additional information

**Competing interests:** The authors declare no competing interests.

