## [Peer Review File · Nature Communications]

Reviewers' comments:

Reviewer #1 (Remarks to the Author):

this article reports TEM study of self-assembly and dynamics of nanoparticles. The liquid state TEM gained a great deal of interest recently, especially stemming from the elegant works of Berkeley group. The soft matter assemblies and dynamics of this kind, to the best of my knowledge have not been explored. The technique could have a strong impact in the studies of nano colloids combined with soft matter systems like micelles, liquid crystals and polymers. The article potentially could meet the standards of Nature Comm, but I am currently somewhat disappointed by two aspects. First, the manuscript is rather observational in nature, without much of quantitative analysis or discussion of deeper insights gained. Authors could analyze interaction forces, diffusion constants, etc. Second, I am not sure what I clearly understand what is revealed that was not already expected - it would be helpful to better articulate this. Gold nanoparticles have been studied in other soft matter systems, like micellar liquid crystals [Nanolett 10, 1347 (2010)]. It would be good for authors to give a broader overview of how this study and technique can impinge on related research areas of soft matter systems involving nanoparticles.

Reviewer #2 (Remarks to the Author):

The manuscript is a collection of beautiful cryoTEM imaging of micelle/dendrimer/gold nanoparticle nanostructures in aqueous suspension. The authors show careful imaging of the thin films as inelastic scattering of the electron beam slowly damages the film by primarily allow solvent to sublime/evaporate in the TEM column. Eventually the film ruptures from the center of the beam at the thinnest spot. With the significant electron density contrast of the gold nanoparticles, the authors calculate speeds of the rupturing with stroboscopic measurements. They also make claims about the presence of no solvent remaining in some films prior to film rupturing. The primary claim is that this type of imaging will be an important tool for monitoring/measuring thin film rupturing physics and/or film reorganization.

While I find the cryoTEM gorgeous, the manuscript is primarily a methods paper. While the potential for measurement of fundamental film physics is clear, the current manuscript only shows film retraction speeds after beam damage. In other words, serendipitous removal of solvent will, of course, cause a solvent film to rupture. If the claim is that this type of imaging could reveal film physics, then I would encourage the authors to perform some fundamental measurements on a known system/bulk system that doesn't rely on the not-well-controlled removal of solvent to cause the film disruption. Or, if solvent removal is an interesting point, the technique should reveal things of fundamental importance with respect to the film and not simply rates of the hole growth after the film begin to break. The authors make claims about the film being completely polymer/dendrimer based prior to rupture with contrast observations. This type of claim would be much easier with a different sample that did not start with a solution of particles. This claim somehow would need to be verified further through other measurements. It would be much easier claim to make if the film was, in fact, not a solution but actually a polymer film where solvent removal through beam damage is not a requirement in the technique.

A minor point: the dendrimers with gold nanoparticles encapsulated do not "self-organize" within the thin films in the centers of the grid holes. Rather, the dendrimers complexed with the PMAA block of the PMAA-PEO block copolymers are forced together due to the surface tension of the solvent and thickness of the films. Some may consider this simple semantics. However, "self-organization" or self-

assembly" implies no need for a template, field, or film thickness to organize. As the authors mention, this phenomenon of nanostructures organizing when in confinement of a thin film is a useful tool in cryoTEM to determine nanostructure and solution/solvated structure.

To emphasize, the imaging is beautiful and shows clearly the hole formation, growth, and gold nanoparticle movement during beam damaging of the solvent film. The manuscript is primarily a technique paper that may motivate others to adopt stroboscopic cryoTEM to perform thin film physics measurements.

Reviewer #1 (Remarks to the Author):

this article reports TEN study of self-assembly and dynamics of nanoparticles. The liquid state TEM gained a great deal of interest recently, especially stemming from the elegant works of Berkeley group. The soft matter assemblies and dynamics of this kind, to the best of my knowledge have not been explored. The technique could have a strong impact in the studies of nano colloids combined with soft matter systems like micelles, liquid crystals and polymers. The article potentially could meet the standards of Nature Comm, but I am currently somewhat disappointed by two aspects. First, the manuscript is rather observational in nature, without much of quantitative analysis or discussion of deeper insights gained. Authors could analyze interaction forces, diffusion constants, etc.

We thank the reviewer for recognizing the potential of our work. In the original manuscript, we already quantified the nanoexplosion speed as well as the solvent evaporation rate upon stroboscopic illumination. We did want to take care not to over-analyze the data and limited ourselves to identify the factors playing a role leading to the observed nanoscale thin film rupture as well as the thinning of the superstructure-containing thin film.

We do understand the reviewers interest and request for additional quantification so we have taken up the challenge to do so, although we had to keep in mind we are limited in the data images regarding time resolution. We anyways now also looked into visualizing the global movements of dendrimicelles not only in the final explosion step (in the original ms.), but also during the whole image acquisition series. Hereto, we developed a Particle Image Velocimetry (PIV) script that provides speed and direction of the micelles and nanoparticles between the different stroboscopically acquired frames. We have added the new Figures S12 and S17 (see also end of this document) which, interestingly, show that micelles located toward the edge of a superstructure overall migrate more in between frames, and that those located in the center migrate less, if at all. The new Figure S12 hence illustrates that the nanoexplosion (as also presented in Figure 1) starts from the center of the superstructures and moves radially outward over the whole stroboscopic series acquired. Please see also the response to reviewer 2.

Unfortunately, the framerate at which we can acquire data with our set up currently limits us in the quantification of the migration velocities of the micelles and the micelle embedded nanoparticles, i.e. discrimination of the two. The developments in cryoTEM hardware and image acquisition speed are fortunately fast so we are currently establishing new collaborations to investigate these thin film dynamics with higher time-domain resolution on transmission electron microscopes that allow for high-speed imaging (e.g., Titan/Krios TEMs equipped with direct electron counting cameras that allow for cryoTEM imaging framerates over 1000 fps.) to proceed with this pioneering study.

Second, I am not sure what I clearly understand what is revealed that was not already expected - it would be helpful to better articulate this. Gold nanoparticles have been studied in other soft matter systems, like micellar liquid crystals [Nanolett 10, 1347 (2010)]. It would be good for authors to give a broader overview of how this study and technique can impinge on related research areas of soft matter systems involving nanoparticles.

It is known that extended electron beam irradiation damages cryoTEM samples, however, the dynamic nature of dendrimicelle cores, and their shrinkage under these conditions is completely unexpected and unprecedented. We identified and quantified the various stages during-, and leading to the observed nanoexplosion, and identified the main factors leading up to the observed nanoexplosion.

Our work shows that —contrary to expectations— micellar thin films can still be dynamic under cryoTEM conditions, and show liquid-like behavior. More importantly, our observations show that the dendrimicelle core is dynamic: The gold nanoparticles that are embedded in the dendrimicelle core act as tracer, and their migration during the exposure series reveals —for the first time— that the dendrimicelle core is dynamic with a clear observed clustering of the gold nanoparticles.

We have now updated the manuscript to emphasize more the unique and unprecedented observation of the nanoparticles movement and clustering. We also added the PIV analysis of all the stroboscopic frames. Finally, we have put our observations in a broader context with other research areas such as particle alignment in micellar systems, nanomedicinal applications, catalysis and sensors.

Reviewer #2 (Remarks to the Author):

The manuscript is a collection of beautiful cryoTEM imaging of micelle/dendrimer/gold nanoparticle nanostructures in aqueous suspension. The authors show careful imaging of the thin films as inelastic scattering of the electron beam slowly damages the film by primarily allow solvent to sublime/evaporate in the TEM column. Eventually the film ruptures from the center of the beam at the thinnest spot. With the significant electron density contrast of the gold nanoparticles, the authors calculate speeds of the rupturing with stroboscopic measurements. They also make claims about the presence of no solvent remaining in some films prior to film rupturing. The primary claim is that this type of imaging will be an important tool for monitoring/measuring thin film rupturing physics and/or film reorganization.

While I find the cryoTEM gorgeous, the manuscript is primarily a methods paper. While the potential for measurement of fundamental film physics is clear, the current manuscript only shows film retraction speeds after beam damage. In other words, serendipitous removal of solvent will, of course, cause a solvent film to rupture.

We thank the reviewer for recognizing the elegance and potential of our work toward studying fundamental thin film physics and dynamics. We would like to point out that — although it might indeed be expected that removal of solvent from a solvent film finally causes it to rupture— the embedded nanoparticle function as tracer to visualise the local and global migration dynamics. Furthermore, even if expected, we provide the first clear stroboscopic thinning of the biconcave cyoTEM sample with the concomitant observation of movements inside the film. As a matter a fact, our study not only shows the global migration of dendrimicelles during the beam exposure, but also shows the local migration and clustering of individual nanoparticles inside the dendrimicelle cores, see Figure S20 (see also end of this document).

If the claim is that this type of imaging could reveal film physics, then I would encourage the authors to perform some fundamental measurements on a known system/bulk system that doesn't rely on the not-well-controlled removal of solvent to cause the film disruption. Or, if solvent removal is an interesting point, the technique should reveal things of fundamental importance with respect to the film and not simply rates of the hole growth after the film begin to break.

As mentioned above, the interesting fundamental aspects of our observations are that the micellar thin film that we form, still show dynamics under cryogenic conditions on both the global (superstructure) and local (individual nanoparticles inside a micelle) scale. As elaborated also in the response to reviewer 1, we have now quantified these dynamics using Particle Image Velocimetry analysis, analysing not only the final nanoexplosion step, as done in the original ms., but all the stroboscopic frames (see also end of this document).

As we are limited by our experimental setup regarding time resolution, we have refrained from over-analyzing our data; however, we actually have been able to observe the velocity vectors to visualize the radial extension of the thin film prior to the explosion, to occur via stress mainly in the outside part whilst the centre remains immobile.

The authors make claims about the film being completely polymer/dendrimer based prior to rupture with contrast observations. This type of claim would be much easier with a different sample that did not start with a solution of particles. This claim somehow would need to be verified further through other measurements. It would be much easier claim to make if the film was, in fact, not a solution but actually a polymer film where solvent removal through beam damage is not a requirement in the technique.

We are not sure we understand the reviewer's suggestion on doing the cryoTEM on a sample containing no particles. It is the nanoparticles that have allowed us to do such a detailed study, and moreover we are not aware of sample preparation protocols that would provide well-defined and identifiable biconcave thin film layers.

A minor point: the dendrimers with gold nanoparticles encapsulated do not "self-organize" within the thin films in the centers of the grid holes. Rather, the dendrimers complexed with the PMAA block of the PMAA-PEO block copolymers are forced together due to the surface tension of the solvent and thickness of the films. Some may consider this simple semantics. However, "self-organization" or self-assembly" implies no need for a template, field, or film thickness to organize. As the authors mention, this phenomenon of nanostructures organizing when in confinement of a thin film is a useful tool in cryoTEM to determine nanostructure and solution/solvated structure.

We agree with the reviewer that the terms self-organization and self-assembly imply that no external template or field is required. We have now updated these terms in the manuscript to include the templating aspect; in fact, in our ACS Nano 2017 paper, we explicitly discussed the templated assembly of the dendrimicelle thin films.

To emphasize, the imaging is beautiful and shows clearly the hole formation, growth, and gold nanoparticle movement during beam damaging of the solvent film. The manuscript is primarily a technique paper that may motivate others to adopt stroboscopic cryoTEM to perform thin film physics measurements.

We thank the reviewer again for the nice words and we indeed hope others will study thin film dynamics; we are currently setting up new collaborations to exploit higher time-domain resolution equipment to do so.

*Figure S12) Particle Image Velocimetry analysis results of the generation 9-based superstructure as shown in Figure S8. A window size of 64*64 pixels, corresponding to ~180*180 nm was used, with a spacing of 32 pixels. For clarity reasons, every other velocity vector is plotted.*

*Figure S17) Particle Image Velocimetry analysis results of the generation 9-based superstructure as shown in Figure S13. A window size of 64*64 pixels, corresponding to ~180*180 nm was used, with a spacing of 32 pixels. For clarity reasons, every other determined vector was plotted.*

Figure S20) cryoTEM micrographs (cropped from Figure S8) of a single dendrimicelle under extended electron beam exposure, revealing not only the global migration of dendrimicelles, but also indicating the local migration of nanoparticles inside dendrimicelles. The dotted oval illustrates the dendrimicelle core shrinkage during the exposure series. Scale bars are 10 nm.

REVIEWERS' COMMENTS:

Reviewer #1 (Remarks to the Author):

Authors responded to my report, mainly not by doing a more quantitative analysis but rather making the case that the novelty is in imaging itself rather than new soft matter physics. Considering that the other referee also seems to have a soft matter background, I therefore think the manuscript should be additionally reviewed by an imaging expert to verify the imaging breakthroughs sufficient for Nature Comm. I can recommend publication if such an imaging expert recommends

Reviewer #2 (Remarks to the Author):

A big thank you to the authors for their revisions to the manuscript and thoughtful answers to my original review. The additional descriptions of the data revealing the dynamic behavior of the dendrimicelles as well as the further quantification of the gold particle dynamics are nice additions to the paper showing more impact of the technique/measurements. The work is worth revealing to a broader audience as a publication in NComm as it is now written. My comments that were not understood by the authors concerned how it would be nice to show measurements of thin films of different materials that did not require thin film formation of nanoparticles (both gold and dendrimicelles) from a solution. In other words, it would be great to show the impact of the technique on bulk/solid films of polymers. Perhaps the authors could mix gold nanoparticles into a desired melt of polymer and perform some type of microtomey to get thin films thin enough for a similar analysis. While they might not be biconcave, they still may exhibit required inelastic interactions with the beam to cause gold nanoparticle movement/rupture. Certainly worth exploring in the future for a broader impact of the work.

REVIEWERS' COMMENTS:

Reviewer #1 (Remarks to the Author):

Authors responded to my report, mainly not by doing a more quantitative analysis but rather making the case that the novelty is in imaging itself rather than new soft matter physics. Considering that the other referee also seems to have a soft matter background, I therefore think the manuscript should be additionally reviewed by an imaging expert to verify the imaging breakthroughs sufficient for Nature Comm. I can recommend publication if such an imaging expert recommends

We thank the reviewer for taking the time reading through our revised manuscript. As mentioned in the response accompanying the revised manuscript, We did want to take care not to over-analyze the data. Therefore, we here limited ourselves to identify the factors playing a role leading to the thinning of the superstructure-containing thin film, as well as the subsequent rupture. In the future, we hope to continue this investigation, and report on it in due time.

Reviewer #2 (Remarks to the Author):

A big thank you to the authors for their revisions to the manuscript and thoughtful answers to my original review. The additional descriptions of the data revealing the dynamic behavior of the dendrimicelles as well as the further quantification of the gold particle dynamics are nice additions to the paper showing more impact of the technique/measurements. The work is worth revealing to a broader audience as a publication in NComm as it is now written.

We thank the reviewer for the kinds word regarding the revised manuscript, and for taking the time reading it.

My comments that were not understood by the authors concerned how it would be nice to show measurements of thin films of different materials that did not require thin film formation of nanoparticles (both gold and dendrimicelles) from a solution. In other words, it would be great to show the impact of the technique on bulk/solid films of polymers. Perhaps the authors could mix gold nanoparticles into a desired melt of polymer and perform some type of microtomey to get thin films thin enough for a similar analysis. While they might not be biconcave, they still may exhibit required inelastic interactions with the beam to cause gold nanoparticle movement/rupture. Certainly worth exploring in the future for a broader impact of the work.

We, once again, thank the reviewer for the provided suggestion, which is clear now, and agree that the provided experiment would be highly interesting to study and we will look into that in the near future.